# Magnetic Fe/Fe_3_C@C Nanoadsorbents for Efficient Cr (VI) Removal

**DOI:** 10.3390/ijms232315135

**Published:** 2022-12-01

**Authors:** Laura Cervera-Gabalda, Cristina Gómez-Polo

**Affiliations:** 1Departamento de Ciencias, Universidad Pública de Navarra, Campus de Arrosadia, 31006 Pamplona, Spain; 2Institute for Advanced Materials and Mathematics (INAMAT^2^), Universidad Pública de Navarra, Campus de Arrosadia, 31006 Pamplona, Spain

**Keywords:** magnetic nanocomposite, thermal decomposition, chromium, adsorption

## Abstract

Magnetic carbon nanocomposites (α-Fe/Fe_3_C@C) synthesized employing fructose and Fe_3_O_4_ magnetite nanoparticles as the carbon and iron precursors, respectively, are analyzed and applied for the removal of Cr (VI). Initial citric acid-coated magnetite nanoparticles, obtained through the co-precipitation method, were mixed with fructose (weight ratio 1:2) and thermally treated at different annealing temperatures (*T_ann_* = 400, 600, 800, and 1000 °C). The thermal decomposition of the carbon matrix and the Fe_3_O_4_ reduction was followed by thermogravimetry (TGA) and Fourier transform infrared (FTIR) spectroscopy, X-ray diffraction, Raman spectroscopy, SQUID magnetometry, and N_2_ adsorption–desorption isotherms. A high annealing temperature (*T_ann_* = 800 °C) leads to optimum magnetic adsorbents (high magnetization enabling the magnetic separation of the adsorbent from the aqueous media and large specific surface area to enhance the pollutant adsorption process). Cr (VI) adsorption tests, performed under weak acid environments (pH = 6) and low pollutant concentrations (1 mg/L), confirm the Cr removal ability and reusability after consecutive adsorption cycles. Physical adsorption (pseudo-first-order kinetics model) and multilayer adsorption (Freundlich isotherm model) characterize the Cr (VI) absorption phenomena and support the enhanced adsorption capability of the synthesized nanostructures.

## 1. Introduction

The environment is clearly affected by numerous anthropometric activities that produce diverse emerging contaminants. These contaminants usually end up in the water resources and air, reducing the availability of clean freshwater sources and affecting air quality [1]. Emerging pollutants in water can be classified into organic pollutants, including pesticides, pharmaceutical products, and care products, as well as inorganic pollutants such as heavy metals (e.g., Pb, Cd, Cr, Co, As, Zn, Hg, and Ni). Specifically, toxic metals discharged primarily from industrial wastewater and mining have become contaminants of emerging concern (CEC) due to their ability to produce adverse effects on both human health and environmental ecosystems.

Among heavy metal pollutants, Cr (VI) stands out due to its high toxicity and mobility. Chromium (Cr) is a transition metal that displays a complex chemistry due to its numerous oxidation states (from +6 to −2). Chromium in wastewater mainly comes from the steel industry, paints, and pigments, wood treatments, or leather tanning. In aqueous media, there are mainly hexavalent Cr (VI) and trivalent Cr (III) [2]. These two Cr species are present as a function of the pH, the redox potential, the total Cr concentration, among other factors. Cr (III) species in aqueous media are found in low concentrations, being its toxicity relatively low [3]. However, Cr (VI) is toxic, causing different diseases such as liver and kidney damage or respiratory problems even in low concentrations [4]. Due to its high toxicity, the limit of total chromium concentration in drinking water is limited to 0.1 mg/L [5].

Currently, different wastewater technologies are employed to remove Cr (VI): adsorption, reduction, precipitation, and ion exchange. However, most of the employed methods for Cr removal are only efficient for high chromium concentrations [6,7]. Physical and chemical adsorption has become one of the most used methods due to its simplicity, low cost, and environmental advantages. Among them, carbon-based adsorbents have been extensively analyzed in the literature and applied in current wastewater applications. The specific microstructure allows the interaction with different pollutants by electrostatic forces, non-covalent forces, or hydrophobic interactions, and different carbon nanostructures (carbon nanotubes, graphene, graphite, and activated carbon) have been widely employed [8]. Along with their high adsorption capacity, associated with a high surface area, the versatility of carbon adsorbents is linked to the possibility to be obtained by different low-cost procedures (e.g., biomass pyrolysis) [9]. However, the decantation and separation of the adsorbent from the aqueous media and its reuse after subsequent treatment cycles represents a current technological issue in the search of cost-effective processes [10]. In this context, magnetic carbon-based nanostructures, i.e., Fe-based nanoparticles surrounded by a carbonaceous matrix, have been proposed as efficient adsorbent agents due to their multifunctionality (combination of a magnetic component and the functional pollutant adsorbent). In these adsorbents, the magnetic nucleus enables the easy separation of the adsorbent from the solution after the wastewater treatment, employing an external magnetic field [11]. High magnetic susceptibility and saturation magnetization, as those of the most employed iron compounds, facilitate the magnetic separation procedure and the reuse and recyclability of the adsorbent.

Several methods have been reported to synthesize iron carbon-based nanostructures: thermal decomposition method, arc-discharge method [12,13], chemical vapor deposition [14], and detonation method [15]. Particularly, the thermal decomposition method stands out due to its simplicity, low-cost reactants, and the possibility to use different types of carbon sources [16]. Generally, Fe^0^ (α-Fe) and/or Fe_3_C nanoparticles immersed in a carbon mesoporous matrix can be obtained from the thermal decomposition of organic matrices employing an Fe^3+^ salt (i.e., Fe_3_Cl 6H_2_O [17], Fe(NO_3_)_3_·9H_2_O [18]. In these synthesis procedures, the salt displays the double role of catalyzing the carbon reduction and simultaneously the role of iron source to form the magnetic nucleus. However, in this work a different strategy, barely addressed in the literature, is followed, namely the reduction of magnetite Fe_3_O_4_ magnetic nanoparticles (MNPs) through thermal treatments under controlled atmosphere, employing fructose (sugar) as a reducing carbon agent [19]. A systematic physicochemical characterization was performed to find the optimum thermal treatment conditions. Fe–C mesoporous adsorbents are obtained, able to efficiently adsorb Cr (IV) in aqueous media for low concentrations (≈1 mg/L) and in weak acid environments (pH = 6). In fact, the adsorption characterization of low pollutant concentrations is scarcely analyzed in previous works, and most of them are performed in acid conditions [20]. Additionally, the reuse and recycling of the magnetic nanoadsorbents was also evaluated, showing the synthesized nanostructures optimum response after subsequent adsorbing cycles.

## 2. Results and Discussion

### 2.1. Structural and Magnetic Characterization

Prior to the analysis of the annealed samples, the decomposition process of the initial MNPs (citric-coated Fe_3_O_4_ nanoparticles) and the mixture with fructose (weight ratio 1:2) were analyzed by TGA under N_2_ atmosphere. Figure 1 shows the weight loss (%) as a function of temperature (*T*), together with the derivative curve, of the analyzed samples. The TGA scan for single fructose is also shown in Figure 1b for comparison. The TGA analysis allows for the determination of the temperatures at which the organic component (citric acid and fructose) starts to degrade, and thus when the carbon graphitic matrix and the Fe_3_O_4_ reduction take place. As Figure 1a shows, the weight loss in the initial MNPs takes place in two main steps (∆*T*_1_ and ∆*T*_2_): ∆*T*_1_ = 150–240 °C and ∆*T_2_* = 340–500 °C. These two decomposition processes, correlated in this case with the decomposition of the citric acid coating, are usually ascribed to the degradation of free or not bounded organic acids at a low temperature (∆*T*_1_) and the final decomposition of the bounded acids at ∆*T*_2_ [21,22].

Similarly, two main steps are detected in the mixture of Fe_3_O_4_ and fructose (∆*T*_1_ = 150–360 °C and ∆*T*_2_ = 390–670 °C). However, some remarkable changes are detected in this sample in comparison with the initial MNPs. First, a highest weight loss at ∆*T*_1_ (46% in comparison with 22% for the initial MNPs), that should be mainly correlated to the contribution of the fructose decomposition. Notice that the fructose decomposes within this temperature range (see Figure 1b). The second decomposition process (∆*T*_2_) not only displays a higher weight loss (23%, double than the value for the initial MNPs), but also an enlargement in the maximum temperature (670 °C, 500 °C for the initial MNPs). This extension in ∆*T*_2_ range may be associated with the further reduction in Fe_3_O_4_ MNPs at high temperatures. Thus, the TGA results conclude that to fully reduce the MNPs mixed with fructose, an annealing temperature higher than 670 °C should be employed.

The decomposition process in the annealed samples as a function of *T_ann_* was first analyzed through FTIR analysis. Figure 2 depicts the comparative FTIR spectra of the fructose and the annealed samples at different *T_ann_* (400, 600, 800, and 1000 °C). The fructose spectrum shows different characteristic functional groups: -OH at 3400–3300 cm^−1^; C_sp3_-H at 2900 cm^−1^; C=O at 1730 cm^−1^; C=C (from the aromatics) at 1620 cm^−1^; C-O in the range of 1400-1060 cm^−1^. The main effect of the thermal treatments is to produce an appreciable reduction in -OH and the carbohydrate bands. Furthermore, the samples annealed at high annealing temperatures also depict the existence of functional groups, compatible with graphitic carbon contributions: C=C stretching vibration (≈1600 cm^−1^), which could be ascribed to the graphitic carbon, and C-O band (≈ 1050 cm^−1^), which could indicate the presence of a graphene oxide structure [23]. In fact, the relative intensity of the corresponding bands of the graphitic carbon slightly increases with *T_ann_* (see Figure 2b).

In order to confirm the decomposition of the organic matrix and the Fe_3_O_4_ reduction, the annealed samples were verified through X-ray diffraction (XRD). Figure 3 shows the XRD patterns corresponding to the graphitic carbon phase (a and c) and to the Fe-based phases (b and d) for two selected annealing temperatures (*T_ann_* = 600 and 800 °C). For *T_ann_* ≤ 600 °C (see Appendix A and Figure 3a), a single diffraction peak appears, which can be linked to a hexagonal *P6_3_mc* space group. However, for *T_ann_* ≥ 800 °C, the carbon peak is clearly asymmetric (see Appendix A and Figure 3c). To properly fit this graphitic contribution, an additional hexagonal (*P6_3_/mmc*) graphite phase was introduced. As previously reported [24], the occurrence of two graphitic planes can be interpreted as the coexistence of two carbon phases with different-order degrees. The interlayer spacing of graphitic carbon can be related to the crystallinity of the carbon phase, a decrease in the *c* parameter being an indication of a higher crystallinity [25]. With respect to the magnetic phases, high temperature annealings (*T_ann_* ≥ 600 °C) lead to the magnetite reduction and the occurrence of Fe_3_C and α-Fe phases (Appendix A and Figure 3b–d). In fact, at *T_ann_* = 400 °C, no clear changes in the initial magnetic phase are observed, all the peaks corresponding to the magnetite phase (Fe_3_O_4_) with similar mean crystallite sizes (≈10 nm) (see Appendix A).

Table 1 summarizes the cell parameters and the relative percentage of the different carbon and Fe phases present in the annealed samples. First, regarding the graphitic phase, the samples annealed at lower temperatures (*T_ann_* ≤ 600 °C) display a value of *c* compatible with a high-ordered crystalline phase (Graphite1). This result could be associated with the occurrence of highly ordered graphitic layers at the first stages of the decomposition process, as it will be later confirmed through Raman spectroscopy. Conversely, for higher annealing temperatures (*T_ann_* ≥ 800 °C), the coexistence of two well-defined graphitic phases can be detected with different interplanar spacings (*c*). In fact, the percentage of the phase with larger crystallinity (lower *c*) is found for *T_ann_* = 1000 °C, indicating an increase in order in the carbon matrix with *T_ann_*.

With respect to the Fe-based phases, for *T_ann_* = 400 °C, the XRD patterns show a single cubic spinel structure (*Fd*$\bar{3}$*m*) space group for Fe_3_O_4_ with *a* = *b* = *c* = 8.358 (2) Å and mean crystallite size 74.58 (4) Å (estimated through Sherrer formula). At *T_ann_* = 600 °C, cementite (Fe_3_C) is the main magnetic phase and for *T_ann_* ≥ 800 °C, the coexistence of Fe_3_C and α-Fe phases is detected with a slight increase in the percentage of α-Fe phase for *T_ann_* = 1000 °C in comparison with *T_ann_* = 800 °C. The transformation of Fe_3_C at high annealing temperatures is compatible with the equilibrium Fe–C phase diagram [26] and with previous studies showing that in a carbon-rich environment at high temperatures, Fe_3_C is unstable and transforms into Fe^0^ and graphitic carbon phases [27]. It is worth noting that this phase transformation has associated a reduction in the mean crystallite size of the cementite phase with *T_ann_*.

Scanning transmission electron microscopy (STEM), together with the corresponding EDX analysis, confirm the previous XRD analysis. Figure 4 shows the STEM images of the samples for *T_ann_* = 400, 800, and 1000 °C and their corresponding size distribution (histogram) using normal (solid red line) or lognormal distribution (dotted black line) functions as fitting functions.

For *T_ann_* = 400 °C (Figure 4a), an ultrafine structure is obtained compatible with the coexistence of Fe_3_O_4_ nanoparticles and surrounded by a highly disordered carbon matrix. The EDX analysis confirms the coexistence of C, O, and Fe in this sample (see Appendix A). Additionally, the calculated average particle diameter, 11 ± 3 nm, is similar to the value of the initial Fe_3_O_4_ nanoparticles. For *T_ann_* = 600 °C (Appendix A), a remarkable increase in the particle dimensions can be detected and the oxygen peak in the EDX analysis almost disappeared, indicating the MNPs reduction. Nevertheless, a small fraction of the sample was not reduced (see the inset of Appendix A) even though it could not be detected by the XRD. Finally, for *T_ann_* = 800 °C and 1000 °C (Figure 4c,d and Figure 4e,f, respectively), nanoparticles with a mean grain diameter higher than 80 nm and surrounded by a carbon matrix are found. Additionally, two size distributions can be distinguished in both samples. Considering the previous XRD analysis (Table 1), the smaller average particle diameter obtained from the histograms could be correlated with the Fe_3_C phase and the larger one with the α-Fe phase.

In order to verify the order state of the carbonaceous matrix in the different annealed samples, Raman spectroscopy was employed. As Figure 5 shows, the Raman spectra can be divided into two main regions: (a) a first-order region (1100–1800 cm^−1^) characterized by the *G* band correlated with the vibrations associated with the graphene and crystalline graphite, and the *D* band linked to the disordered carbon; (b) a second region, where the second-order of the *D* band (2*D*, overtone of the *D* band) appears, together with other overtones (*D* + *D’’*, *D* + *G*, and 2*D’*). For *T_ann_* ≤ 600 °C, a wide disordered 2*D* band is detected. The ordering of the carbon phase for *T_ann_* ≥ 800 °C can be confirmed by the occurrence of additional peaks in the second-order region (2*D’*, *D* + *G* and *D* + *D’’*). Furthermore, for *T_ann_* = 400 °C, an additional peak at *ω* ≈ 550 cm^−1^ is found, linked to the presence of the iron oxides (Fe_3_O_4_) and confirming that in this annealed sample, the reduction process has not taken place.

Table 2 summarizes the fitted Raman parameters employing a Lorentz peak deconvolution (solid lines in Figure 5); the mean Raman peak positions (*ω_D_*, *ω_G_* and *ω*_2*D*_), full width at half maximum (FWHM) for both *D* and *G* bands, and relative peak intensities of the first- and second-order regions (*I_D_*/*I_G_* and *I*_2*D*_/*I_D_*). Although no remarkable variations are found in *ω_D_* and *ω_G_*, a clear decrease is found in *ω*_2*D*_ with *T_ann_* that can be correlated to the carbon-ordering process. In fact, similar results are found in other Fe–C nanostructures obtained from the decomposition of sugars and Fe^3+^ salts [16]; that is, a highly ordered carbon matrix with *ω*_2*D*_ ≈ 2700 cm^−1^ and a shift toward higher wavelengths (*ω*_2*D*_ ≈ 2900 cm^−1^) for the carbon-disordered state. Furthermore, a diminution in FWDM in both peaks, *D* and *G,* with *T_ann_* is detected and associated with an increase in the structural correlation length, *L_a_*, of the graphitic phase (FWHM ∝ 1/*L_a_*) [28]. The detected decrease in FWHM with *T_ann_* may again confirm the increase in the carbon ordering upon annealing.

Moreover, the ratios *I_D_*/*I_G_* and *I*_2*D*_/*I_D_* also reflect the order state of the carbon in the samples (i.e., decrease and increase, respectively, with the increase in the order degree). As Table 2 shows, a noticeable increase in *I*_2*D*_/*I_D_* is found as *T_ann_* increases, so the carbon ordering with the annealing temperature is again confirmed. Furthermore, with the exception of the sample annealed at *T_ann_* = 400 °C, a decreasing trend of *I_D_*/*I_G_* with *T_ann_* can also be concluded. The anomalous low value on this sample (*I_D_*/*I_G_* = 0.69) could be interpreted as a result of the initial ordering steps in the carbon matrix. In fact, previous results show that when amorphous carbon is being ordered to form nanocrystalline graphite, an initial increment in *I_D_*/*I_G_* is first detected [29].

The magnetic characterization of the annealed samples supports the previous conclusions regarding the evolution of the decomposition process with *T_ann_*. Figure 6 shows the magnetic hysteresis loops (*M-H*) at 300 K for the annealed samples. An increase in the high-field magnetization (saturation magnetization, *M_S_*) is obtained as *T_ann_* increases as a consequence of the reduction from Fe_3_O_4_ into Fe_3_C for *T_ann_* = 600 °C and the coexistence of cementite (Fe_3_C) and α-Fe at high annealing temperatures (*T_ann_* ≥ 800 °C). Considering that *M_S_* bulk values are 98 emu/g for Fe_3_O_4_, 140 emu/g for Fe_3_C, and 220 emu/g for α-Fe, it is reasonable to conclude that samples with a large amount of α-Fe phase (as for *T_ann_* = 1000 °C) should display higher magnetization values, in good agreement with XRD Rietveld refinement (Table 1).

Furthermore, for *T_ann_* = 400 °C, the anhysteretic response at 300 K is compatible with the superparamagnetic nature of the nanoparticles. The reduction in Fe_3_C nanoparticles for *T_ann_* = 600 °C promotes the disappearance of the superparamagnetic behavior and non-zero values of the coercive field (*H_c_* = 121, 47 and 13 Oe, for *T_ann_* = 600, 800 and 1000 °C, respectively). Comparing the annealed samples at 800 and 1000 °C, which contain a similar magnetic phase distribution, the decrease in *H_c_* with *T_ann_* should be mainly ascribed to the increase in the crystallite size of the α-Fe phase (see Table 1) and the multidomain nature of the Fe-based nanoparticles [13].

To conclude, those samples annealed at high temperatures (*T_ann_* ≥ 800 °C) display the main characteristics to be employed as efficient magnetic adsorbents (i.e., high magnetic susceptibility and high saturation magnetization).

### 2.2. Cr (VI) Adsorption Tests

The adsorption capacity is determined by the specific surface area and porosity of the adsorbent, among other factors. As the surface area increases, the adsorption sites of the pollutants on the sample surface increase, leading to an improvement on the adsorption capacity of the samples [30].

Figure 7 shows the nitrogen adsorption–desorption curves of the initial MNPs and the selected annealed samples (*T_ann_* ≥ 800 °C). The annealed samples display a type IV isotherm, while the initial MNPs are characterized by a type V (IUPAC classification [31]). Both behaviors are related to a mesoporous microstructure with a hysteresis attributed to adsorption metastability and network effects. In fact, the three samples display similar pore diameter values (≈ 12 nm) in the mesoporous range. A remarkable increase in the *BET* surface area is found for the annealed sample at 800 °C (383 m^2^/g) in comparison with the initial citric acid-coated MNPs (123 m^2^/g). As a consequence of the carbon ordering, the annealed sample at the highest temperature (*T_ann_* = 1000 °C) displayed a reduced *BET* surface of 77 m^2^/g.

Therefore, in order to analyze the adsorption capacity of the Fe–C system, the sample annealed at 800 °C was selected as optimum adsorbent for the following Cr (VI) adsorption tests. This sample complies with the desired specifications of an optimum magnetic pollutant adsorbent: partially ordered carbon matrix (linked to a high specific surface area) and a high magnetization and magnetic susceptibility to facilitate the magnetic separation of the adsorbent. The maximum adsorbent Cr (VI) capacity of the selected optimum nanocomposite was first analyzed. Figure 8a displays (mean values from three tests) the percentage (%) of Cr (VI) in the aqueous solution (initial 1 mg/L) as a function of the contact time, *t*, between the adsorbent (25 mg) and the Cr (VI) solution (25 mL). The results show a good reproducibility. For comparison, the adsorption curve for the initial MNPs is displayed in the inset of the figure. As it can be seen, the annealed sample, which displays the larger specific surface area, is able to almost fully adsorb (% Cr (VI) ≈ 95%), after few minutes, the Cr anions in the aqueous solution. Furthermore, the occurrence of different functional groups (i.e., hydroxyl (OH-), epoxy (R-O-R), and carboxyl (R-COOH)) [32] linked to the surface of the mesoporous carbon and as a consequence of the sugar decomposition process could enhance the adsorption process in this annealed sample. However, the initial MNPs, despite being covered by an organic and mesoporous coating (citric acid), display a lower adsorption capacity, that is, a maximum Cr (VI) adsorption around 80% for contact times longer than 1 h.

One of the most interesting features of these magnetic adsorbents is the possibility to recover the samples after the adsorption process, employing an external magnetic field and thus enabling their reuse after different adsorption cycles. Accordingly, the reusability of the analyzed adsorbent was evaluated, where after each adsorption cycle the adsorbent was magnetically separated from the aqueous solution by a permanent magnet. Subsequently, the adsorbed Cr (VI) was removed from the samples by employing deionized water and ultrasounds. Then, the cleaned sample was dried at 80 °C and reused for the next cycle. Figure 8b displays % Cr (VI) versus time curves after three adsorption–desorption cycles, where no significant changes in the adsorption curves are found. It is worth noting that it is not necessary to employ alkaline solutions, as in previously reported studies [33,34].

Cr (VI) Adsorption Kinetics

The adsorption kinetics analysis is of great importance, since it provides information about the time needed to reach the equilibrium of the adsorption process. Different kinetics models are usually employed in the literature, the pseudo-first-order and pseudo-second-order models being particularly used. The pseudo-first-order model is correlated with the adsorption process which occurs through diffusion across the interface mainly at the first stages of the adsorption process. On the other hand, the pseudo-second-order model is based on the fact that the limiting step is chemical adsorption and predicts the behavior over the whole range of adsorption [35]. The equations of these two kinetic models can be expressed as:(1)$$\mathrm{Pseudo}-\mathrm{first}-\mathrm{order}:\;\ln(q_{e}-q_{t})=\ln q_{e}-k_{1}t$$
(2)$$\mathrm{Pseudo}-\sec\mathrm{ond}-\mathrm{order}:\;\frac{t}{q_{t}}=\frac{1}{k_{2}q_{e}^{2}}+\frac{t}{q_{e}}\;$$
where *q_t_* and *q_e_* are the amount of pollutant adsorbed (mg_pollutant_/g_adsorbent_) at time *t* (min) and equilibrium, respectively; *k*_1_ is the constant for the pseudo-first-order model (min^−1^) and *k*_2_ the constant for the pseudo-second-order model (g/(mg·min)). The rate constants *(k*_1_ and *k*_2_) provide information about the time required for reaching the equilibrium of the adsorption process within each mode.

Table 3 shows the results of the fitting of the adsorption kinetics of Cr (VI) of the annealed sample employing the pseudo-first-order and pseudo-second-order models. The adsorption kinetic fits better with the pseudo-first order model which corresponds to a physical adsorption process (higher correlation coefficient, *R*^2^, see Figure 9a). It should be noted that most of the reported kinetic constants of similar carbon magnetic adsorbents display lower values (less than one order of magnitude) [36,37,38,39], confirming the optimum adsorption performance of the synthesized magnetic composite. For comparison, the results for the initial MNPs are also included (see Table 3). In this case, a slightly better fitting is obtained employing the pseudo-second-order model that may indicate a predominant chemical adsorption process, although some contribution from physical adsorption processes cannot be completely excluded (see Appendix A). Nevertheless, this sample displays kinetic constant values, one order of magnitude lower that the annealed sample (MNPS + fructose), confirming the optimization of Cr (VI) adsorption with the reduction procedure (carbon coating).

b.Cr (VI) Adsorption Isotherms

In order to describe the interaction between adsorbate and adsorbent, adsorption isotherm models are often used. These isotherms can provide information about the adsorbent capacities, the adsorption phenomena, and the surface properties of the adsorbent. Different models can be employed, and among them Langmuir and Freundlich models stand out. The Langmuir model assumes a homogeneous surface, the adsorption process is performed in a monolayer process, with no lateral interaction between adsorbed molecules and reversible adsorption. On the other hand, the Freundlich model is based on a heterogeneous surface, multilayer and reversible adsorption [35]. These two models are properly described by the following expressions:(3)$$\mathrm{Langmuir}\;\mathrm{model}:\;\frac{c_{e}}{q_{e}}=\frac{C_{e}}{q_{max}}+\frac{1}{q_{max}K_{L}}$$
(4)$$\mathrm{Freundlich}\;\mathrm{model}:\;\ln q_{e}=lnK_{F}+\frac{lnC_{e}}{n}$$
where *q_e_* (mg_pollutant_/g_adsorbent_) corresponds to the amount of the Cr (VI) adsorbed at equilibrium time, *q_max_* (mg_pollutant_/g_adsorbent_) is the maximum adsorption capacity, *C_e_* (mg_pollutant_/L_solution_) is the Cr (VI) concentration at the equilibrium, *K_L_* (L/mg) is the Langmuir constant, and *K_F_* (mg/g·(L/mg)^1/n^) and *n* are Freundlich constants. Additionally, a dimensionless constant *R_L_* is employed to explain the adsorption characteristics of the Langmuir isotherm, being:(5)$$R_{L}=\frac{1}{1+K_{L}C_{e}}$$

For *R_L_* values in the 0–1 range, the adsorption can be considered favorable; *R_L_* = 0 is irreversible; *R_L_* > 1 is unfavorable; and *R_L_* = 1 indicates a linear adsorption.

The fitting parameters of the two employed isotherm models are summarized in Table 4. As in the previous kinetic analysis, both samples display a different response. First, the Freundlich model better describes the adsorption isotherms of the annealed sample (see Figure 9b), with similar values of the fitting parameters (*K_F_*) as those reported in magnetic nanocomposites [38,40]. However, for the initial MNPs, although the Langmuir model describes the adsorption isotherm slightly better, a complex behavior can be inferred looking at the low values of the correlation coefficient *R^2^* for both models (see Appendix A). Nevertheless, the calculated *R_L_* factor (<1) and the high values of *n* (>1) indicate a favorable adsorption process of Cr (VI) in both samples. The higher values of *q_max_* (Langmuir model) and *K_F_* (Freundlich model) for the annealed sample in comparison with those displayed by the initial MNPs again confirm the optimization of the adsorption capacity under the performed synthesis process.

Based on the above analysis, it can be concluded that the mean mechanism governing the Cr (VI) adsorption of the selected Fe–C nanocomposite is the physical adsorption (pseudo-first-order model) and multilayer/reversible process at various active sites (Freundlich model). The electrostatic interaction between the porous carbon surface and the chromate anions may play a dominant role in the adsorption process of the magnetic nanocomposite. This mechanism (electrostatic interaction) may facilitate Cr removal after the adsorption tests and the reuse of the adsorbent after different cycles. However, the initial citric acid MNPs show a different adsorption performance, where the chemisorption (pseudo-second-order model) and monolayer process (Langmuir model) may mainly contribute. In this case, the redox reactions between Cr and the functional groups in the coated MNPs may mainly contribute, although a complex adsorption process can be inferred.

Finally, Table 5 compares the obtained results of the Cr (VI) adsorption analysis with other reported adsorbents. As it is shown, the analyzed Fe–C adsorbent displays a lower maximum adsorption capacity (*q_max_*), but equivalent *K_F_* isotherm constant, than most of the reported adsorbents. However, its larger kinetic constants (*k*_1_ or *k*_2_), close to one order of magnitude higher than the values in most of the reported studies, are worth noting. This fact, together with its simple and low-cost synthesis procedure, and its magnetic nature that allows for magnetic separation, endows this type of nanocomposites with the optimal properties to be used as efficient Cr (VI) adsorbents.

## 3. Materials and Methods

### 3.1. Synthesis of Fe–C Nanostructures

Citric acid-coated Fe_3_O_4_ nanoparticles (MNPs) were synthesized by the co-precipitation method (AIN, Asociación de la Industria Navarra (Cordovilla, Navarre, Spain)). The organic coating was introduced to stabilize the magnetic nanoparticles and avoid their aggregation. First, deionized water was deoxygenated for 30 min with argon flow and was then heated to 60 °C and stirred mechanically (300 rpm). FeCl_2_·4H_2_O and FeCl_3_·6H_2_O were added (2:1 molar ratio). Then, the temperature was increased to 90 °C, under stirring at 1100 rpm and 24 mL of aqueous ammonia was added to induce the precipitation of the nanoparticles. The solution was stirred for 30 min. Finally, 10 mL of citric acid was added dropwise and the mixture was stirred for 60 min. The solution was cooled down to room temperature in an argon atmosphere for 45 min. The samples were centrifuged and washed with deionized water.

To obtain the Fe–C nanostructures, the citric-coated MNPs were mixed with fructose in a mortar–pestle (weight ratio 1:2). Then, the mixture was thermally treated at different temperatures (*T_ann_* = 400, 600, 800, and 1000 °C) in Ar atmosphere employing a heating rate of 10 °C/min until the annealing temperature was reached, this temperature being kept during 60 min. The samples were cooled down inside the furnace until room temperature.

### 3.2. Structural and Magnetic Characterization

The thermal decomposition of the initial citric acid coating on the single MNPs and the mixture MNPs and sucrose was followed by thermogravimetric (TGA) analysis, (HI-RES 2950 TA Instruments) employing a heating rate of 10 °C/min under nitrogen atmosphere. Fourier transform infrared (FTIR) spectroscopy, with KBr as the window material used in pellets, were employed to confirm the changes in the organic components upon thermal treatments. Regarding the structural analysis of the annealed samples, X-ray powder diffractometry, XRD, (Siemens D-5000) with monochromated Cu Kα1 radiation (λ = 1.54056 Å) were employed, using the Rietveld method and the Fullprof program in the analysis of the spectra [45]. Scanning transmission electron microscopy (STEM) with energy dispersive X-ray (EDX) spectroscopy analyses were performed by using a FEI Tecnai Field Emission Gun operated at 300 kV. Raman spectroscopy (Jasco NRS-3100 dispersive Raman spectrophotometer using a 532 nm laser (7 mW) and a 600-line grating covering the range 260–3900 cm^−1^) was used to analyze the evolution of the carbon in the annealed samples. Powder samples, without further preparation, were exposed 0.1 sec per scan and at least 500 scans were accumulated in order to obtain a good signal-to-noise ratio. The specific surface area of the materials can be determined by the Brunauer—Emmett—Teller (*BET*) method, employing a Micrometrics Gemini V (model 2365) instrument that allowed for the determination of N_2_ adsorption–desorption isotherms at 77 K. A SQUID magnetometer (Quantum Design MPMS XL7) was employed to magnetically characterize the samples.

### 3.3. Cr (VI) Adsorption Tests

Cr (VI) adsorption was evaluated employing a UV–Vis spectrophotometer (UV–16, LAN OPTICS) with a wavelength range from 190 nm to 1100 nm. Since aqueous Cr (VI) solutions have no absorption in the UV-Vis range, it is necessary to use an indirect concentration measurement procedure employing a color reagent [46]. This color reagent was prepared by the dissolution of 150 mg of 1,5-diphenylcarbohydrazide into 25 mL of methanol. Then, 125 mL of H_2_SO_4_ aqueous solution, prepared with 7 mL of H_2_SO_4_, was added to the above solution. Finally, the mixture was diluted with D.I. water to 250 mL. The Cr (VI) reacted with the 1,5-diphenylcarbohyzazide, forming a colored (pinkish) complex, absorbing at a wavelength of 540 nm. Calibration curves were initially performed to properly evaluate the concentrations of the Cr (VI) solutions. Two different concentration ranges were calibrated, employing aqueous solutions with different Cr (VI) concentrations using K_2_Cr_2_O_7_ salt. Then, 1 mL of the Cr (VI) aqueous solution was mixed with 2 mL of the color reagent solution, forming the colored complex. Finally, the absorbance, *A*, of the mixture was measured. The results provide the following calibration parameters: Concentration (mg/L) = *A*/0.184 (up to 100 mg/L of Cr (VI); Concentration (mg/L) = *A*/0.225 (up to 1 mg/L of Cr (VI)). For each test (pH = 6), 25 mL of Cr (VI) water solution (1 mg/L Cr (VI)) with 25 mg of the adsorbent was prepared. Then, the solutions were mechanically stirred and aliquots of 1 mL were collected at different times. The adsorbent was separated by a hand-held magnet, and filtered by a 0.22 μm syringe filter. The adsorption kinetics were analyzed employing pseudo-first-order and pseudo-second-order models, analyzing the evolution of the amount of pollutant adsorbed as a function of time. Additionally, the Cr (VI) adsorption isotherms were determined, preparing aqueous solutions with different Cr (VI) concentrations (from 0.2 to 100 mg/L) with the same amount of adsorbent (1 mg/mL). The solutions were stirred by a shaker (600 rpm) overnight. Then, the adsorbent was separated by a hand-held magnet and the supernatant was filtered by a 0.22 μm syringe filter. Langmuir and Freundlich models were employed to evaluate the adsorption isotherms of the studied samples.

## 4. Conclusions

The synthesis procedure of magnetic carbon nanocomposites, composed by Fe-based nanoparticles surrounded by a carbon matrix, have been analyzed, enabling the design of efficient magnetic nanoadsorbents. The thermal decomposition of fructose employing the catalytic effect of Fe of magnetite nanoparticles provides an optimum mesoporous nanostructure for the removal of Cr (VI) in aqueous solutions. In fact, the performed structural and magnetic characterization enables us to conclude that, under high annealing temperatures (800 °C), it is possible to obtain optimum magnetic adsorbents, with high magnetization linked to the reduction in the initial Fe_3_O_4_ phase to α-Fe and Fe_3_C; and large specific surface area of the carbon matrix leading to enhanced Cr adsorption. The optimized adsorbent is able to efficiently remove Cr in weak acid environments (pH = 6) and at low pollutant concentrations (1 mg/L) under short contact times (minutes). Its reusability is enhanced by the possibility to adequately separate it from the aqueous media by the action of an external magnetic field. Physical adsorption (pseudo-first-order kinetics model) and multilayer adsorption (Freundlich isotherm model) characterize the Cr (VI) absorption phenomena, where electrostatic interactions between adsorbate and adsorbent may mainly contribute to the optimum response after consecutive cycles.

## Figures and Tables

**Figure 1 ijms-23-15135-f001:**
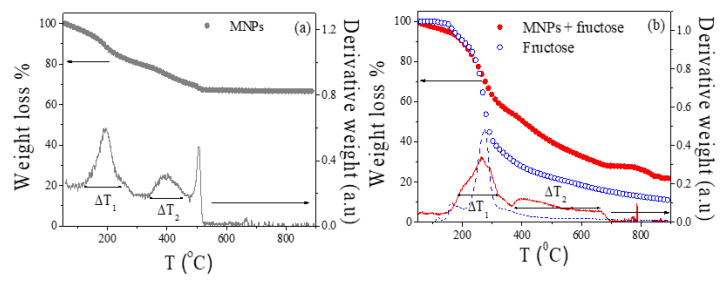
TGA scans for the (**a**) initial Fe_3_O_4_ MNPs and (**b**) (•) MNPs mixed with fructose. TGA scan for (o) fructose is included for comparison. The derivative of the TGA curves (lines) are also shown (dashed line for the initial fructose).

**Figure 2 ijms-23-15135-f002:**
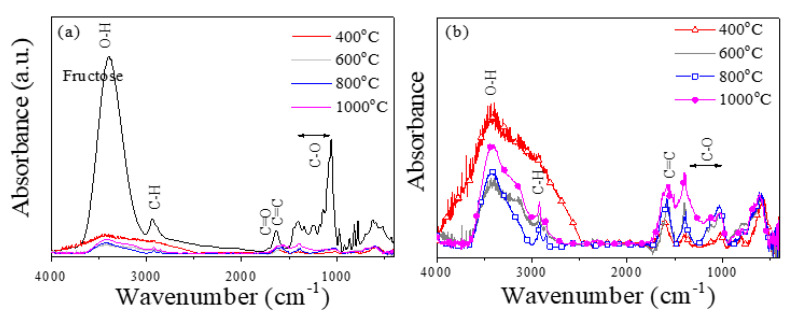
FTIR spectra of the (**a**) MNPs + fructose samples annealed at different temperatures. Fructose spectrum is also included for comparison. (**b**) Enlargement of the spectra of the annealed samples.

**Figure 3 ijms-23-15135-f003:**
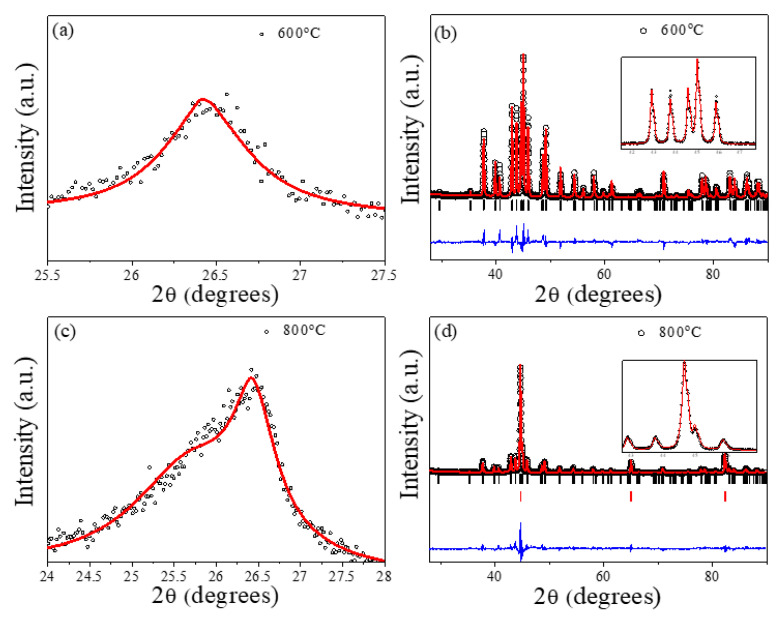
XRD patterns for the MNPs + fructose samples annealed at different temperatures: (**a**) and (**b**) *T_ann_* = 600 °C: (**c**,**d**) *T_ann_* = 800 °C. (o) Experimental, (—) calculated (Rietveld refinement) intensities and (—) difference between both intensities. The Bragg reflections are shown for (∣) Fe_3_C and (∣) α-Fe. Insets: enlargement for 42 ≤ 2*θ* ≤ 48 region.

**Figure 4 ijms-23-15135-f004:**
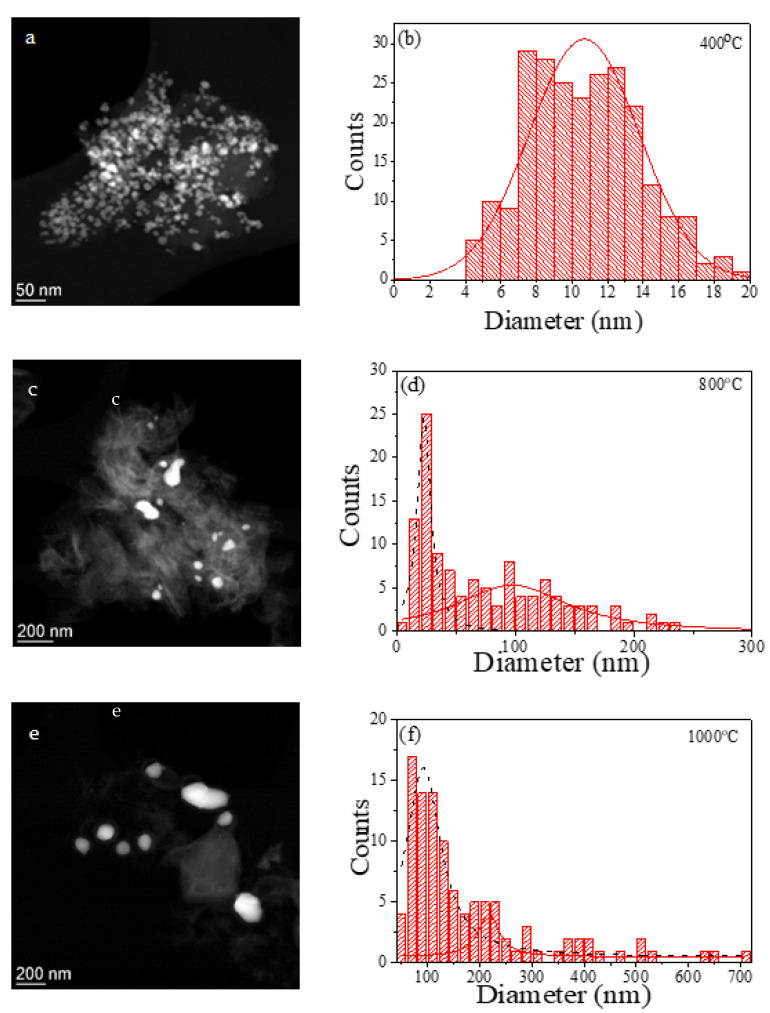
STEM images of the annealed samples at (**a**) 400 °C, (**c**) 800 °C, and (**e**) 1000 °C, and their corresponding histograms (**b**,**d**,**f**).

**Figure 5 ijms-23-15135-f005:**
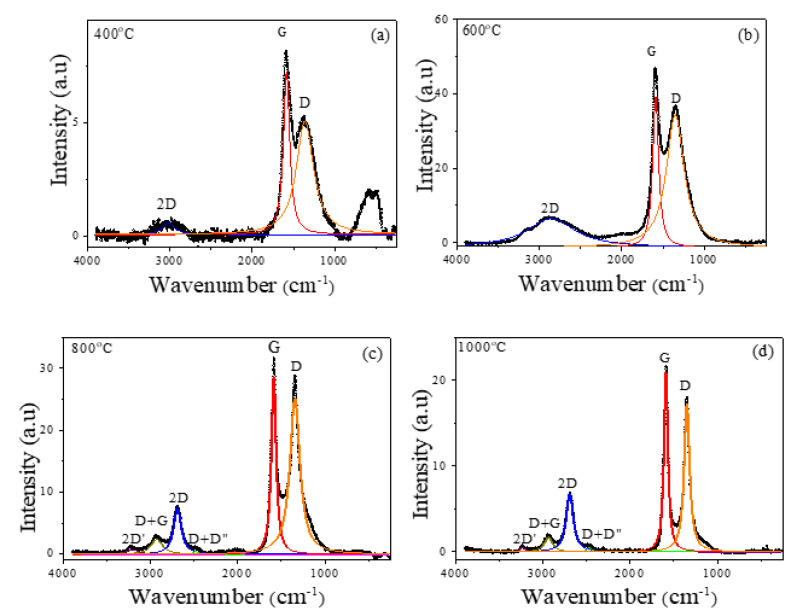
Raman spectra of the annealed samples at (**a**) 400 °C, (**b**) 600 °C, (**c**) 800 °C, and (**d**) 1000 °C. Solid lines: Lorentz peak deconvolution.

**Figure 6 ijms-23-15135-f006:**
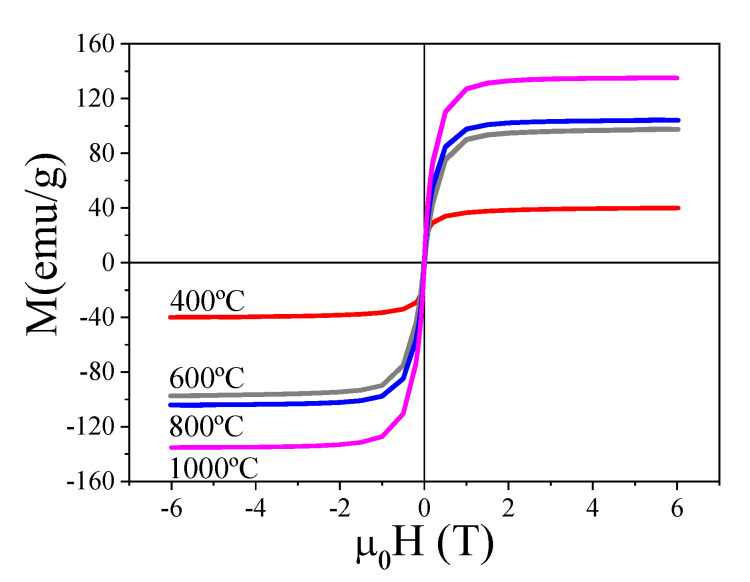
Magnetic hysteresis loops at 300 K of the annealed samples.

**Figure 7 ijms-23-15135-f007:**
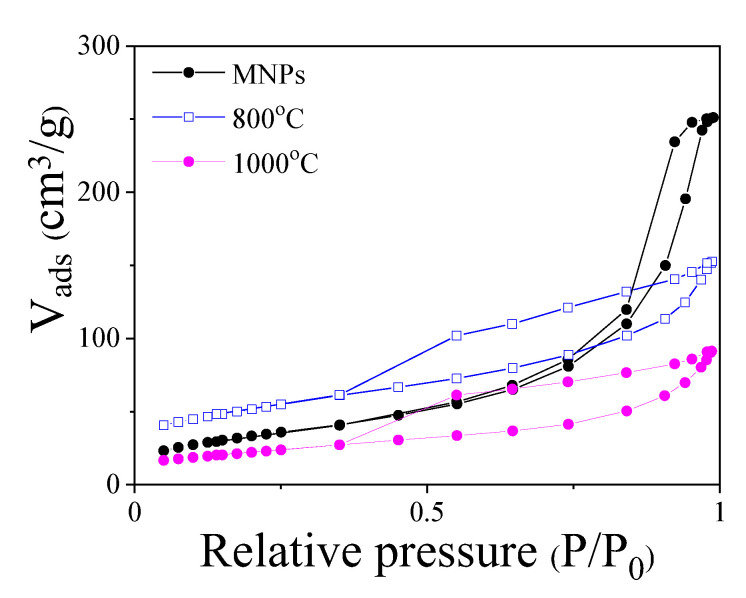
Nitrogen adsorption–desorption curves at 77 K for the MNPs and the annealed sample (MNPs + fructose) at 800 °C and 1000 °C.

**Figure 8 ijms-23-15135-f008:**
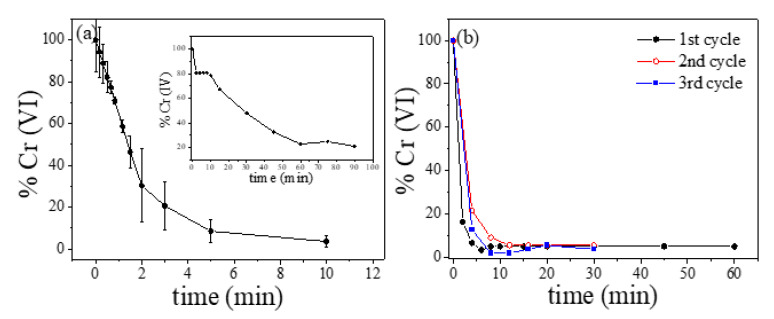
(**a**) Cr (VI) adsorption (% Cr (VI)) in the presence of the selected annealed sample. Inset: initial citric acid-coated MNPs. (**b**) Reusability adsorption tests (adsorption curves after 3 subsequent cycles).

**Figure 9 ijms-23-15135-f009:**
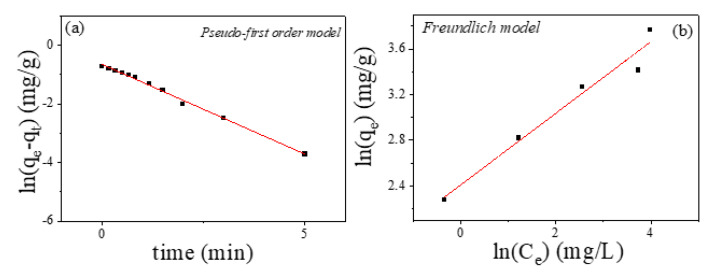
(**a**) Adsorption kinetics of Cr (VI) and (**b**) adsorption isotherms employing the selected annealed sample.

**Table 1 ijms-23-15135-t001:** Parameters obtained from Rietveld refinement of the XRD of the graphitic and Fe-based phases.

*T_ann_*	Graphitic Phase	Cell Parameter (Å)	Fe-Based Phases	Cell Parameter (Å)	Crystallite Size (Å)
400 °C	Graphite1 (100%)	*a* = *b* = 2.454 (1) *c* = 6.696 (6)	Fe_3_O_4_ (100%)	*a* = *b* = *c* = 8.3619 (6)	79.14 (4)
600 °C	Graphite1 (100%)	*a* = *b* = 2.439 (8) *c* = 6.742 (2)	Fe_3_C (100%)	*a* = 5.0920 *b* = 6.7441 (1) *c* = 4.5269	777.45 (8)
800 °C	Graphite1 (31.2%)	*a* = *b* = 2.446 (2) *c* = 6.741 (1)	Fe_3_C (61.1%)	*a* = 5.0926 (1) *b* = 6.745 (2) *c* = 4.5281 (1)	603.32 (5)
	Graphite2 (68.8%)	*a* = *b* = 2.557 (2) *c* = 6.932 (5)	α-Fe (38.9%)	*a* = *b* = *c* = 2.8682 (1)	689.76 (0)
1000 °C	Graphite1 (70.7)	*a* = *b* = 2.5681 (6) *c* = 6.7478 (6)	Fe_3_C (57.8%)	*a* = 5.0934 (1) *b* = 6.7456 (2) *c* = 4.5290 (1)	542.91 (2)
	Graphite2 (29.3)	*a* = *b* = 2.3812 (9) *c* = 6.869 (3)	α-Fe (42.2%)	*a* = *b* = *c* = 2.8682 (1)	931.48 (5)

**Table 2 ijms-23-15135-t002:** Raman-fitted parameters for samples annealed at different temperatures.

*T_ann_*	*ω_D_* (cm^−1^)	*ω_G_* (cm^−1^)	*ω*_2*D*_ (cm^−1^)	FWHM_D_ (cm^−1^)	FWHM_G_ (cm^−1^)	*I_D_*/*I_G_*	*I*_2*D*_/*I_D_*
400 °C	1362 ± 1	1585 ± 1	3018 ± 10	253 ± 5	92 ± 2	0.69	0.10
600 °C	1351.3 ± 0.3	1589.6 ± 0.1	2859 ± 3	262 ± 1	86.3 ± 0.4	0.88	0.22
800 °C	1344.6 ± 0.3	1587.6 ± 0.1	2692 ± 1	139 ± 1	63.9 ± 0.4	0.88	0.29
1000 °C	1350.4 ± 0.1	1590.1 ± 0.1	2693.7 ± 0.3	75.8 ± 0.3	53.7 ± 0.2	0.83	0.39

**Table 3 ijms-23-15135-t003:** Kinetic parameters for adsorption of Cr (VI) in the presence of the annealed sample (*T_ann_* = 800 °C) and initial MNPs.

Sample	Pseudo-First-Order Model			Pseudo-Second-Order Model		
	*q_e_* (mg/g)	*k*_1_ (min^−1^)	*R* ^2^	*q_e_* (mg/g)	*k*_2_ (g/mg min)	*R* ^2^
MNPs + fructose (*T_ann_* = 800 °C)	0.519	0.61 ± 0.01	0.994	0.611	0.72 ± 0.06	0.887
MNPs	0.746	0.047 ± 0.005	0.909	0.937	0.042 ± 0.004	0.928

**Table 4 ijms-23-15135-t004:** Adsorption isotherms parameters for adsorption of Cr (VI) in the presence of the annealed sample (*T_ann_* = 800 °C) and initial MNPs.

Sample	Langmuir Model				Freundlich Model		
	*q_max_* (mg/g)	*K_L_* (L/mg)	*R_L_*	*R* ^2^	*K_F_* (L/g)	*n*	*R* ^2^
MNPs + fructose (*T_ann_* = 800 °C)	41.5	0.182	0.499	0.907	11.1	3.18	0.955
MNPs	11.2	0.709	0.940	0.869	4.85	5.99	0.724

**Table 5 ijms-23-15135-t005:** Comparison of maximum Cr (VI) adsorption capacity and kinetic constants with reported adsorbents.

Sample	Isotherm Values	Kinetic Constants	References
MNPs + fructose (*T_ann_* = 800 °C)	*q_max_* = 41.5 mg/g	*k*_1_ = 0.61 min^−1^	This work
	*K_F_* = 11.1 L/g	*k*_2_ = 0.72 g/mg·min	
Poly(ionic liquid)	*q_max_* = 236.8 mg/g	*k*_2_ = 0.0211 g/mg·min	[41]
Hierarchical polydopamine coated cellulose nanocrystal microstructures	*q_max_* = 205 mg/g	*k*_1_ = 0.0028 min^−1^	[42]
	*K_F_* = 43.27 L/g	*k*_2_ = 3.96 × 10^−4^ g/mg·min	
Hierarchical porous polydopamine microspheres	*q_max_* = 307.7 mg/g	*k*_1_ = 2.508 min^−1^	[43]
	*K_F_* = 94.99 L/g	*k*_2_ = 3.39 g/mg·min	
Ti-PDA nanoparticles	*q_max_* = 625 mg/g	*k*_1_ = 0.01 min^−1^	[44]
	*K_F_* = 0.28111 L/g	*k*_2_ = 2.45 × 10^−3^ g/mg·min	

## Data Availability

Not applicable.

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
