# Peer review of "Magnetic Fe/Fe3C@C Nanoadsorbents for Efficient Cr (VI) Removal"

_ijms, 2022, doi:10.3390/ijms232315135_

Round 1

Reviewer 1 Report

This manuscript developed magnetic carbon nanocomposites (α-Fe/Fe3C@C) synthesized employing fructose and Fe3O4 magnetite nanoparticles as the carbon and iron precursors and explored for the removal of Cr (VI). It exhibited good Cr(VI) removal performance at weak acidic conditions and recyclability. It can be accepted after the following concerns to be addressed.

1. The morphologies of the prepared magnetic carbon nanocomposites should be carefully characterized via TEM and SEM.

2. For practical application, it is important to know the effect of other coexisting ions on the Cr(VI) removal performance. Please add it.

3. To state its advantage over the reported Cr(VI) adsorbents, a table or figure to compare its performance to other adsorbents would be preferable, such as Chem. Eng. J., 2019, 378, 122107.,Cellulose 2017, 26 (11), 6401-6414; Journal of Leather Science and Engineering 2020, 2 (1), 1-10; , Appl. Surf. Sci., 2022, 580, 152168; Appl. Catal. B Environ., 2019, 251, 229-239.

Author Response

Please, see attached document.

Reviewer 2 Report

The paper is interesting and well written, the author addresses the problem of Cr removal from the environment (mainly wastewater) using adsorbents such as magnetic carbon-based nanostructures. The Fe-based nano-particles surrounded by a carbonaceous matrix, are used due to their combination of a magnetic component and the functional pollutant adsorbent.

The authors present a method for the reduction of magnetite Fe3O4 magnetic nanoparticles (MNPs) through thermal treatments under controlled atmosphere, employing fructose as reducing carbon agent. A systematic physicochemical characterization was performed to find the optimum thermal treatment conditions. Fe-C mesoporous adsorbents are obtained, able to efficiently adsorb Cr in aqueous media for low concentrations (≈ 1 mg/L) and under weak acid environment. They are using different techniques to study and evaluate the thermal decomposition of the carbon matrix and the Fe3O4 reduction such as: Thermogravimetry (TGA), Fourier-Transform Infrared spectroscopy (FTIR), X-ray diffraction, Raman spectroscopy, SQUID 15 magnetometry and N2 adsorption-desorption isotherms.

Abstract is well synthesized, emphasizing the main achievements of authors. The introduction is comprehensive and provides information about the current stage and the trends in this field of research. Anyway I suggest the addition of new references about chrome oxidation states (+6 to -2), line 38, in the second paragraph and in the line 78 in the fourth paragraph of the introduction.

Results and Discussions section is comprehensive and well organized, the figures and tables are clear. However I consider that the authors should specify the type of atmosphere used in TGA measurements, this can be useful for other scientists working in the field.

 The Conclusions are clear and highlights the contribution of the authors regarding a new method for the reduction of magnetite nanoparticles through thermal treatments under controlled atmosphere, employing fructose as reducing carbon agent.

Overall, the paper is of good quality and in my opinion deserve to be published after minor revision.

Author Response

Please, see attached document.

Round 2

Reviewer 1 Report

The authors have addressed all my concerns